# Collateral Vessels Have Unique Endothelial and Smooth Muscle Cell Phenotypes

**DOI:** 10.3390/ijms20153608

**Published:** 2019-07-24

**Authors:** Hua Zhang, Dan Chalothorn, James E Faber

**Affiliations:** Department of Cell Biology and Physiology, Curriculum in Neuroscience, McAllister Heart Institute, University of North Carolina, Chapel Hill, NC 27599-7545, USA

**Keywords:** collateral circulation, cerebral collaterals, endothelial cells, ischemic stroke, primary cilia

## Abstract

Collaterals are unique blood vessels present in the microcirculation of most tissues that, by cross-connecting a small fraction of the outer branches of adjacent arterial trees, provide alternate routes of perfusion. However, collaterals are especially susceptible to rarefaction caused by aging, other vascular risk factors, and mouse models of Alzheimer’s disease—a vulnerability attributed to the disturbed hemodynamic environment in the watershed regions where they reside. We examined the hypothesis that endothelial and smooth muscle cells (ECs and SMCs, respectively) of collaterals have specializations, distinct from those of similarly-sized nearby distal-most arterioles (DMAs) that maintain collateral integrity despite their continuous exposure to low and oscillatory/disturbed shear stress, high wall stress, and low blood oxygen. Examination of mouse brain revealed the following: Unlike the pro-inflammatory cobble-stoned morphology of ECs exposed to low/oscillatory shear stress elsewhere in the vasculature, collateral ECs are aligned with the vessel axis. Primary cilia, which sense shear stress, are present, unexpectedly, on ECs of collaterals and DMAs but are less abundant on collaterals. Unlike DMAs, collaterals are continuously invested with SMCs, have increased expression of *Pycard, Ki67, Pdgfb, Angpt2, Dll4, Ephrinb2*, and eNOS, and maintain expression of *Klf2/4*. Collaterals lack tortuosity when first formed during development, but tortuosity becomes evident within days after birth, progresses through middle age, and then declines—results consistent with the concept that collateral wall cells have a higher turnover rate than DMAs that favors proliferative senescence and collateral rarefaction. In conclusion, endothelial and SMCs of collaterals have morphologic and functional differences from those of nearby similarly sized arterioles. Future studies are required to determine if they represent specializations that counterbalance the disturbed hemodynamic, pro-inflammatory, and pro-proliferative environment in which collaterals reside and thus mitigate their risk factor-induced rarefaction.

## 1. Introduction

Obstructive disease, i.e., stroke and atherosclerosis of the coronary and peripheral arteries, is the leading cause of morbidity and mortality. Collateral circulation, which is composed of anastomotic vessels called collaterals, is the most important system capable of mitigating the effects of obstructive disease [1,2,3,4,5]. However, the number and diameter of collaterals in tissues vary greatly among individuals for reasons that are only beginning to be understood [1]. Moreover, little is known about the basic biology of collaterals, in part because of their small diameter and low density in tissues, difficulty in distinguishing them from other vessels, and lack of methodologies allowing study of their endothelial cells (ECs) and smooth muscle cells (SMCs) in cell culture. It is known, however, that collaterals are unique with regard to their: mechanism of formation during development (collaterogenesis) and its high sensitivity to genetic background-dependent variation, anatomic location in the circulation, hemodynamic forces acting on their ECs and SMCs, hallmark tortuosity, high susceptibility to risk factor-induced rarefaction, robust shear stress-dependent outward remodeling, and protective function in ischemic disease [1,2,3,4,5].

In mice, collaterals form late during gestation by the sprouting of ephrin-B2^+^ ECs off of a small number of distal arterioles that subsequently undergo a tip cell-led proliferation, migration, and lumenization process to establish anastomoses between adjacent arterial trees [6,7,8]. This process varies greatly due to naturally-occurring polymorphisms in certain genes in the collaterogenesis pathway, resulting in wide differences in collateral extent and thus collateral-dependent blood flow and tissue injury in experimental models of occlusive arterial disease [9,10,11]. Collateral blood flow also varies widely in humans suffering stroke [3,4], coronary, and peripheral artery diseases [12,13], with recent evidence supporting involvement of the same key variant gene identified in the mouse [14]. Since collaterals interconnect adjacent arterial trees, there is little or no pressure drop across them in the absence of obstructive disease. Thus, blood flow within collaterals is near zero, oscillating slowly toward one or the other tree that they anastomose [6,15]. Endothelial cells and SMCs that compose the collateral wall are therefore continuously exposed to low and disturbed shear stress, high circumferential wall stress, and low blood oxygen content. Accompanying this adverse hemodynamic environment, which favors vascular inflammation and endothelial dysfunction elsewhere in the circulation, collaterals in mice undergo a progressive decline in number and a loss of diameter with aging, presence of vascular risk factors, and in models of Alzheimer’s disease [16,17,18,19,20,21]. Supportive evidence, based on measurement of collateral-dependent flow induced by acute ischemic stroke, has been reported in humans with aging and presence of vascular risk factors [22,23,24]. This so call “collateral rarefaction” has been linked in mice to chronic low-level inflammation, endothelial dysfunction, and increased proliferation of collateral ECs and SMCs [16,17,18,19,20,25]. Collaterals are also capable of undergoing robust anatomic outward remodeling in steno-occlusive disease, compared to similarly sized arterioles [2,5,9,10]. Additionally, different from arterioles in the trees that they interconnect, collaterals lack myogenic responsiveness and have less SMC tone at baseline [26,27].

Despite these important unique features of collaterals, no studies have examined cellular and molecular aspects of their endothelial and smooth muscle cells to determine whether they differ from those of similarly sized arterioles. For example, blood flow in arteries and arterioles of healthy tissues, and thus fluid shear stress experienced by their ECs, is laminar, orthograde, and high velocity, with the exception of the inner curvature of the aortic arch and sites immediately downstream of arterial bifurcations where flow varies from high-velocity and orthograde to low-velocity with transient flow reversals in a fraction of the fluid laminae during each cardiac cycle (i.e., “disturbed” flow) [28,29,30,31]. Endothelial cells at these sites of disturbed shear stress are cobblestone in shape, as opposed to elsewhere where they are elongated and aligned with the vessel axis. They also evidence higher levels of proliferation, apoptosis, permeability, lipid uptake, oxidative stress, and markers of inflammation and aging, i.e., displaying the well-known pro-atherogenic EC phenotype specific to these sites. Flow and shear stress in collaterals in the absence of obstruction share similarly disturbed conditions [6,15]. Yet this is the normal environment in which collateral ECs and SMCs reside.

The purpose of this study was to examine the hypothesis that collateral ECs and SMCs express unique morphological and functional phenotypes that serve as adaptations or specializations to maintain the integrity of the collateral wall and balance against collateral rarefaction favored by the low and disturbed shear stress, high wall stress, and low blood oxygen levels—the latter caused by low flow-induced hemoglobin unloading to tissue. Such differences, if present, could offer insights into how collaterals are able to persist in healthy individuals but also why they are so susceptible to rarefaction with aging and other vascular risk factors. In addition, knowledge concerning the molecular features of collateral wall cells is fundamental to better understand how collaterals form during development, remodel in obstructive disease, and undergo risk factor-induced rarefaction, and to investigate possible treatments to augment or intervene with these processes.

## 2. Results

### 2.1. Collateral Endothelial Cells Are Aligned with the Vessel Axis Despite their Chronic Exposure to Low and Oscillatory Shear Stress

As a first-test of the hypothesis that collateral wall cells have unique phenotypes, we assessed the orientation of collateral ECs using scanning electron microscopy (SEM) and staining of the junctional protein zona occludens-1 (ZO-1). Despite the unique low and disturbed shear stress present in collaterals, ECs of collaterals were aligned with the vessel axis to the same extent, i.e., ~4 degrees off the longitudinal axis, which was present in nearby distal-most arterioles (DMAs) and in the descending thoracic aorta where blood flow is high-velocity and laminar (Figure 1 and Figure 2). The area, perimeter, length, and width of collateral ECs were also comparable to ECs lining DMAs (Appendix A).

### 2.2. Endothelial Cells of Collaterals and Distal-Most Arterioles Have Primary Cilia; Collaterals Have Fewer

While examining EC orientation using SEM we noticed the presence of casts of channels penetrating into a fraction of the ECs lining collaterals (Figure 3). Based on studies in other cell types including bovine aorta and human umbilical vein ECs [32,33,34], these are invaginations of the plasmalemma that abut the ciliary membrane to form the ciliary pocket, which houses the proximal end of luminal primary cilia (PrC) that were removed by shearing forces (“depilated/de-ciliated”) during infusion of Batson’s #17. The Batson’s-filled ciliary pockets (i.e., PrC) were commonly located near the nucleus, in accordance with the base of the cilium being associated with the basal body [34,35,36,37,38,39,40]. Distal arterioles also have cilia (Appendix A). One, two, or rarely three cilia were present in ECs that expressed them (Figure 3, Appendix A). More than one cilium per EC has not been found in previous studies, to our knowledge; however, previous reports have only examined large conduit vessels. The above method, which serendipitously identified PrC, is indirect since ciliary pockets were what were detected. We therefore confirmed their presence using immunofluorescence (Figure 4). Eighteen percent of collateral ECs expressed PrC, as compared to 28% of DMAs; thus ECs lining collaterals had 34% fewer cilia. We have not found other reports that ECs within vessels of the microcirculation express primary cilia.

### 2.3. Collaterals Are Invested with a Continuous Layer of Smooth Muscle Cells, unlike Distal-Most Arterioles Whose Smooth Muscle Cells Are Discontinuous

Smooth muscle cells (SMCs) become sparse and discontinuous on distal arterioles in many tissue types [41,42,43]. Unlike arterioles that have orthograde flow, collateral blood flow converges from opposite directions, resulting in an average flow of near or at zero in the center-most segment of the collateral in the absence of occlusion (Figure 2A). The kinetic energy of flow is therefore converted to potential energy, which increases the circumferential wall stress of collaterals. Accordingly, we reasoned that SMC investment of collaterals might be greater than that present on DMAs to balance this increased wall stress. Figure 5 supports this hypothesis.

### 2.4. Gene Expression Differs for Collaterals Versus Distal-Most Arterioles

Given the disturbed hemodynamic, pro-oxidative stress (i.e., low blood oxygen content) environment in which collateral mural cells reside, plus the high susceptibility of collaterals to undergo rarefaction with aging, vascular risk factor presence, and eNOS/NO deficiency [16,17,18,19,20,21,25] compared to nearby DMAs [17], we postulated that expression of genes involved with inflammation, cell proliferation, aging, and angiogenesis differ for collaterals and DMAs. To test this hypothesis, we measured transcript levels of 22 such genes (Figure 6), as well as eNOS immunofluorescence (Figure 7). Collaterals had increased expression of *Pycard, Ki67, Pdgfb, Angpt2, Dll4, Ephrinb2*, and eNOS, whereas 16 other genes were not significantly different. Of note, expression of the EC shear stress-sensitive transcription factors Klf2 and Klf4, which promote anti-proliferative, -inflammatory, and -angiogenic processes, upregulate eNOS, and are sharply downregulated at vascular sites of low and disturbed shear stress [35,36,37,44,45], were, in contrast, not downregulated in collaterals compared to DMAs with laminar high-velocity flow.

### 2.5. Changes in Tortuosity over a Collateral’s “Lifetime” Suggests Accelerated Proliferative Senescence of their Mural Cells

We previously reported that aging, hypertension, and other vascular risk factors cause a loss of collateral number and a smaller diameter in those that remain [17,18,19,25]. Aging-induced rarefaction becomes evident in late middle-age in mice (16 months of age), is not seen in DMAs, is accelerated in onset by the presence of other vascular risk factors, and is associated with increased cellular markers of oxidative stress, inflammation, proliferation, and aging, as well as increased vessel tortuosity [17,18,19,25]. Tortuosity is a hallmark of collaterals. We postulated that it arises from a persistently increased rate of proliferation of collateral wall cells (confirmed in [18]) that is driven by the disturbed hemodynamic conditions present in collaterals. The above studies did not examine tortuosity at time points earlier than 3 months of age. To determine whether collaterals begin to acquire tortuosity from formation onward, we examined mice on embryonic day E15.5 (collaterals form between E15.5 and E18.5 [6,7]) and on post-natal day-1 and day-21. Tortuosity was absent at E15.5 but became evident by P1 and was significant by P21 (Figure 8). When viewed in context with our previous data for tortuosity at later ages (Figure 9; human year-equivalents for the latter five bars are approximately 13, 19, 49, 69, 84 years [47]), these findings support the hypothesis that collateral rarefaction, which becomes evident in late middle-age in mice [17], is caused by proliferative senescence of collateral ECs and SMCs due to a lifetime elevation in proliferation in excess of apoptosis that is caused by the disturbed hemodynamic conditions in which collaterals reside. 

## 3. Discussion

The present study identified a number of distinct features of collateral ECs and SMCs that may underlie or contribute to the unique characteristics of collaterals outlined in the Introduction. We found that despite the low and oscillatory shear stress present in collaterals at baseline in the absence of obstruction, their ECs are aligned with the vessel axis to the same degree present in distal-most arterioles and the descending aorta—vessels with high-velocity orthograde laminar flow. Endothelial cells of both collaterals and arterioles have primary cilia, and collaterals possess fewer of them. Smooth muscle cells of collaterals are continuous, unlike those of DMAs. Collaterals have higher levels of expression of genes associated with both pro- and anti-inflammatory and pro- and anti-proliferation pathways, compared to DMAs. A bias towards a higher rate of cell proliferation in collateral mural cells [18] is supported by the observation that collaterals begin to acquire tortuosity shortly after their formation in the embryo that increases through middle age (16 months, 49 human year-equivalents [47]). The above findings provide insights into structural and molecular specializations that may underlie the unique features and functions of collateral vessels.

Convergence of blood flow in collaterals at baseline imposes low and “disturbed” flow/shear stress forces on their mural cells, i.e., either absence of flow or very low flow that oscillates to-and-fro (~1–10 times per minute) and averages zero [6,15]. This results in increased wall stress according to the Bernoulli relationship. When present elsewhere in the arterial circulation, e.g., at bifurcations, the inner arch of the aorta or downstream of plaques, low and disturbed shear stress favors a non-aligned cobblestoned EC morphology that is associated with increased oxidative stress, inflammation, markers of aging, and low eNOS/NO activity (i.e., endothelial dysfunction) [28,29,30,31,44,45]. Surprisingly, however, we found that collateral ECs have the same alignment (and cell dimensions) as ECs in DMAs and the descending aorta. We did not investigate how this anti-inflammatory structural phenotype is specified. It is possible that one or more of the other unique features that we identified are involved (see below). On the other hand, the data in Figure 2 for flow and shear stress were obtained in anesthetized animals. During awake behaviors, collaterals may have periods of sustained flow in one or the other direction induced by changes in regional metabolic activity in the territory supplied by the arterial trees that they cross-connect. It is not known if collaterals contribute in this way to physiological metabolic regulation of blood flow and oxygen delivery, i.e., neurovascular coupling in brain and functional hyperemia elsewhere. However, such periods of sustained unidirectional flow could promote the EC orientation we observed. Irrespective of underlying cause, we speculate that the aligned phenotype of collateral ECs is part (or a marker) of a group of protective mechanisms that favor maintenance of collaterals and mitigate their rarefaction (Figure 9). 

To our knowledge, this is the first report showing that ECs lining collaterals and arterioles have primary cilia. They are much more abundant than reported previously for conduit vessels from healthy individuals (i.e., absent or present on less than 1% of ECs [32,34]): 18% of collateral ECs have PrC compared to 28% for DMAs. Primary cilia on ECs were described in 1984 by Haust in aorta of rabbits and humans with atherosclerosis [48]. Subsequent reports described ciliated ECs in capillaries of the pineal gland of a 20 week-old human fetus [49], developing heart and aortic valve leaflets [33,50,51], and references therein], surrounding atheromas [52], ectopically in the common carotid artery of *ApoE*^−/−^ mice, and in healthy individuals at bifurcations of conduit arteries and the inner curvature of the aortic arch; by contrast cilia are absent or nearly so in regions of arteries where shear stress is laminar [33,40,50]. Most other cell types express PrC during development, under certain conditions of cell culture [53], and in adults [35,36,37]. Depending on cell type, PrC participate in specification of embryo asymmetry, centriole disposition, proliferation/cell cycle regulation, autophagy, flow-sensing mechanotransduction, chemoception, and compartmentalization and trafficking of signaling proteins among the cilioplasm, cytoplasm, and nucleoplasm (e.g., for Gli and PDGFRα) [35,36,37]. When present, cilia are most abundant in non-proliferating cells, with the proximal end anchored in an invagination of the plasmalemma (ciliary pocket) in ECs and other but not all cell types [33,34,36]. Primary cilia are complexed with the mother centriole of the basal body that links to the microtubule-organizing center (MTOC) [32,35,36,37,54]. Disassembly/resorption of the cilium during S-phase and centriole liberation are essential for cell division [36,37,54]. Since PrC are linked to the cytoskeleton via the MTOC, flow-induced ciliary bending in ECs [55] is capable of being transmitted throughout the cell, including to cell–cell and cell–matrix junctions [33,37]. Primary cilia transduce fluid shear stress in renal tubular epithelial cells and ECs through a pathway that that is exceptionally sensitive to shear stress and involves polycystin-1 and polycystin-2, which are encoded by *Pkd1* and *Pkd2* [35,36,37]. Polycystin-1 has mechanosensitive properties while polycystin-2 is a TRP calcium channel. Both proteins are required to sense shear stress and in turn release nitric oxide [33,37]. Defects in PrC are associated with many abnormalities. For example, mutations of *PKD1* and *PKD2* are causal for autosomal dominant polycystic kidney disease, with kidney ECs and tubular cells of patients evidencing deficient calcium and NO responses and increased proliferation [33,35,36,37]. 

Primary cilia are absent in human umbilical vein ECs maintained under laminar shear stress and proliferative quiescence in cell culture. In human umbilical veins less than one percent of ECs have cilia that protrude into the lumen, while in a larger fraction the cilia are located intracellularly [34,55]. Embryonic aorta and ECs cultured from it have a single cilium that projects into the lumen [38,39,40]. Endothelial cilia are present in regions of high shear stress during embryonic development, along with expression of the shear stress-sensitive transcription factor, KLF2, which transactivates *eNOS* and other anti-inflammatory and anti-proliferative genes [44,45]. In regions with low or disturbed shear stress PrC are disassembled/absent and expression of *Klf2* and *eNOS* are abolished and reduced, respectively [33,56]. Expression of *Klf2* is also inhibited in non-ciliated ECs isolated from embryonic arteries, and chemical removal of PrC from ECs in culture has a similar effect, i.e., abolishing *Klf2* expression [57]. Interestingly, in ECs of adult *ApoE*^−/−^ mice, which have endothelial dysfunction but do not develop plaques, cilia are expressed ectopically in the common carotid artery despite the presence of laminar flow, compared to wildtype mice that are devoid of cilia [33]. Cilia were lost when high shear stress was induced via implantation of a flow-restricting cast around the vessel. Casting of common carotids in wildtype mice induced ciliogenesis only in regions of low and disturbed shear stress. These findings suggest that expression of PrC on ECs in vivo in adults is limited to regions of low/disturbed shear stress but can occur ectopically in arteries with laminar flow in the presence of endothelial dysfunction caused by hyperlipidemia [33] and possibly other vascular risk factors. Of note, in highly ciliated, disturbed-flow areas such as the inner curvature of the aortic arch or downstream of plaques, approximately 25% of ECs have a single cilium while the rest are devoid [33], a percentage similar to what we observed in arterioles and collaterals. 

Our findings that PrC are also present on arterioles and collaterals in healthy young adult mice highlight the need for studies examining cilia function in these vessel types. This includes determining whether our observation of fewer cilia on collateral ECs than arterioles has functional significance. Endothelial cells are coupled mechanically, electrically, and diffusionally to adjacent ECs and SMCs [58], thus only a fraction of ECs may need to express cilia for transduction of mechano-sensitive or other signals. High shear stress causes disassembly of cilia in cultured ECs, while oscillatory flow reversal induces their expression [33]. We speculate that cilia on arteriole and collateral ECs may reflect the lower flow/shear stress in arterioles and very low and disturbed flow in collaterals and that fewer PrC on collateral ECs may serve to reduce their sensitivity to the prevailing disturbed shear stress environment. In other words, fewer cilia on collateral ECs may be part of a repertoire of adaptations that balance against or oppose—through maintained or increased expression of KLF2/4, eNOS, and other anti-inflammatory/anti-proliferative factors—the low-grade inflammatory, oxidative, proliferative, and apoptotic signals promoted by the disturbed hemodynamic environment present in collaterals (Figure 9). Egorova et al. [36] proposed something similar, i.e., that since presence of PrC on ECs is associated with KLF2 expression, endothelial cilia may signal a brake on EC activation in regions of low and disturbed flow. A protective role for cilia in these regions [36,37] is supported by the recent report that removal of endothelial cilia using conditional deletion of *Ift88* increased atherosclerosis and inflammatory gene expression, and decreased eNOS activity in *Apoe*^−/−^ mice fed a high-fat diet [59], and that the endothelium becomes sensitized in athero-prone regions to undergo osteogenic differentiation in Tg737 (*orpk/orpk*) cilium-defective mice [60]. It is also possible that if PrC are protective, fewer of them in collaterals could contribute to the high susceptibility of these vessels to rarefaction from aging and other vascular risk factors. However, fewer cilia on collaterals might simply reflect a secondary or bystander effect, being a consequence, for example, of a higher inherent proliferation rate of collateral ECs as evidenced by the progressive increase in collateral tortuosity (discussed below), since presence of PrC and their association with the basal body is believed to favor removal of cells from the cell cycle [35,36,37]. Future studies will be required to determine if our finding of multiple cilia on ECs reflects ECs that have undergone proliferative senescence and associated failed cytokinesis and nuclear polyploidy [39]. 

It has recently been shown that ECs in the developing mouse retina rely on PrC to stabilize vessel connections during remodeling of the vascular plexus in regions with low to intermediate shear stress [61]. Endothelial cilia sense flow in zebrafish embryos, participate in recruitment of mural cells to arterial fated vessels, and are required for normal vascular morphogenesis [62,63]. The number and diameter of collaterals declines beginning in middle age [17]. This age-induced rarefaction is strongly accelerated by genetic or pharmacologically induced eNOS/NO deficiency or the presence of vascular risk factors [16,19]. Increased shear stress induces collateral outward remodeling following acute or slowly developing arterial obstruction [1,2,5]. *Pkd1*^+/−^ mice and patients with autosomal dominant polycystic kidney disease have endothelial eNOS/NO dysfunction [64]. It will be important to examine in future studies whether collateral PrC participate in one or more of the above functions using EC-specific knockdown of polycystin-1, since: 1) deficiency in it leads to altered ciliary function, 2) polycystin-1 together with polycystin-2 participate in flow-sensing by PrC, 3) mutant forms of either protein cause polycystic kidney disease [35,36,37], 4) there is evidence that VHL, independent of its role in the degradation of Hif1α, together with GSK3β are required for structural maintenance of the cilium [65], and 5) the protein Rabep2, which is required for collaterogenesis [11], is a novel substrate of GSK3β [66], localizes at the cilium–basal body complex, and knockdown of it leads to defective ciliogenesis [67]. Other approaches to interfere with cilia presence and function, e.g., with knockdown of other ciliary proteins such as *Pkd2* and *Ift88*, also will need to be examined. 

In contrast to distal arterioles, which have sparse and discontinuous SMCs in various tissues including retina (we were unable to identify studies in brain) [41,42,43], SMCs were continuous on collaterals. We speculate this may be an adaptive increase in wall thickness to balance the increase in circumferential wall stress caused by the Bernoulli-specified conversion of kinetic energy of flow to increased potential energy (transmural pressure) as a consequence of flow-convergence in collaterals. It would be interesting to examine whether the composition and amount of extracellular matrix, which could assist SMCs in balancing the increased wall stress in collaterals, differs in collaterals versus arterioles. Of note, despite their increased SMC coverage, collaterals have less rather than more tone compared to similarly-sized arterioles, and lack myogenic responsiveness—additional unique features of collateral vessels [26,27].

The disturbed hemodynamic, pro-oxidative environment in which collateral mural cells reside led us to examine whether expression of genes involved in inflammation, cell proliferation, aging, and angiogenesis differ for collaterals versus distal arterioles. Collaterals displayed increased mRNA levels for the pro-inflammatory, pro-apoptosis inflammasome gene, *Pycard*, the pro-proliferative genes, *Ki67, Pdgfb*, and *Angpt2*, the anti-proliferative gene, *Dll4*, and the differentiated arterial-type EC marker gene, *Ephrinb2*. However, expression of cell cycle inhibitor genes, *p21, p27*, and *p53*, were not different, nor were other genes associated with proliferation, cell cycle arrest, and aging (*p16^Ink4a^, Ampk, Sirt1, telomerase*). Neither were there differences in expression of other genes associated with EC and/or SMC proliferation (*Vegfa, Flk1, Clic4, Pdgfa, Flt1*) and that are required (in the case of the first three genes [7,8]) for formation of collaterals during development or involved with specifying EC and SMC differentiation and quiescence (*Tgfb, Angpt1*). Increased expression by collaterals of the above pro-proliferation genes is consistent with our tortuosity measurements, which suggest that collateral mural cells have a higher proliferation rate, compared to other arterial vessels: collateral tortuosity was evident by the first day after birth, continued to increase through middle age, and then declined. The latter occurred at the same time that collaterals experience a decline in number and diameter with advanced aging [17]. These findings support the hypothesis that age-associated collateral rarefaction is caused by proliferative senescence and subsequent apoptosis of collateral ECs and SMCs due to a lifetime elevation in proliferation rate caused by the disturbed hemodynamic and low blood oxygen content environment in which collaterals reside (Figure 9). 

Collaterals also displayed increased activity of eNOS, which previous studies have shown opposes rarefaction of collaterals caused by aging and other vascular risk factors [16,18,19]. eNOS-derived NO inhibits oxidative stress, inflammation, proliferation, leukocyte adhesion, platelet aggregation, and cellular aging and promotes SMC relaxation [58,68]. Since shear stress is a proximate stimulus for eNOS-derived NO, increased eNOS/NO in collaterals with their low and disturbed shear stress environment may lessen the effect of factors promoting collateral rarefaction (Figure 9). Likewise, maintained expression of the EC shear stress-sensitive transcription factors, *Klf2* and *Klf4*, in collaterals despite their low and oscillatory flow, which inhibits expression of these factors elsewhere in the arterial vasculature with disturbed flow [44], may act as additional “balancing” factors or collateral specializations, along with increased eNOS, aligned ECs, fewer cilia, robust SMC coverage, and increased ephrin-B2 and Dll4. Expression of KLF2 and KLF4, which negatively regulate proliferation, inflammation, and angiogenesis, upregulate eNOS, and are sharply downregulated at sites of low and disturbed shear stress, were not different in collaterals versus DMAs. Interestingly, PrC promote *Klf2, Klf4*, and *eNOS* expression [35,36,37,44,69,70].

A limitation of the above studies is that RNA was obtained from dissected vessels, which are composed of ECs, SMCs, and, although less so, pericytes, fibroblasts, and resident myeloid cells. Studies are needed that employ separation of cell types and examination of a wider array of genes and their respective protein levels. However, the difficulty in manually dissecting collaterals and distal arterioles in the required numbers, the effect of cell dissociation techniques on baseline levels of RNA and protein, the absence of cell culture models of “collateral” ECs and SMCs, and the as of yet lack of any collateral-specific marker gene, preclude the use of these approaches. Of note, however, expression of several of the genes examined are specific or enriched for ECs, e.g., *Flk1, Angpt1, Angpt2, Ephrinb2, DLL4, eNOS, Clic4, Klf2*, and *Klf4*. However, analysis of gene transcription does not always reflect changes in protein level or function; thus the investigation of oxidative stress, inflammatory, proliferation, or senescence markers at the protein level may better reflect differences in cellular characteristics.

## 4. Materials and Methods

Three to 4 month-old male C57BL/6 (B6) wildtype (lab colony, Jackson Laboratories breeders, Bar Harbor, ME, USA) or genetically modified mice were studied. Scanning electron microscopy was performed on the cerebral arterial vasculature that was filled with Batson’s #17. Angiography and morphometry were performed after filling with yellow Microfil^®^ [18]. Prior to infusion of the casting agents, and also for immunohistochemistry, the vasculature was cleared of blood and perfusion-fixed after maximal dilation with 10^−4^ M nitroprusside. Filling was confined to the pre-capillary vessels by adjusting the viscosity and input pressure of the casting material. All pial collaterals between the anterior cerebral artery (ACA) and middle (MCA) trees of both hemispheres were identified; tortuosity (axial length of the collateral ÷ scalar length connecting the collateral’s endpoints [17]) and lumen diameter, EC orientation, and other morphometrics were obtained at midpoint for each collateral and an average value obtained for each animal unless indicated otherwise in the figure legends. Permanent MCA occlusion was by electro-cautery occlusion of the M1-MCA just distal to the lenticulostriate branches [18]. Right femoral artery (FA) occlusion was achieved by ligating the superficial FA immediately proximal and distal to the superior epigastric artery, which was also ligated [18]. Hindlimb perfusion was measured by laser Doppler perfusion imaging of the plantar foot and the adductor thigh regions. Values reported are means ± SE, with significance at *p* < 0.05; *n*-sizes (number of animals studied) and statistical tests are given in the figure legends.

All applicable international, national, and/or institutional guidelines for the care and use of animals were followed, including the National Institutes of Health Guide for the Care and Use of Laboratory Animals (UNC IACUC #18-123.0-A, 1 April 2019). This article does not contain any studies with human participants performed by any of the authors.

### 4.1. Angiography and Morphometry

As previously described in detail [20] animals were anesthetized deeply with ketamine and xylazine and heparinized. The distal thoracic aorta was cannulated, right atrium perforated, and the mouse was perfused with PBS containing freshly prepared sodium nitroprusside (10^−4^ M, for maximal dilation) and Evans blue dye (for light staining of brain and the endothelial surface) at ~100 mmHg. One ml of yellow Microfil (Flowtech Inc, Carver, MA, USA) was infused at a viscosity adjusted to fill the entire pial arterial and collateral circulations with sufficient pressure (~100 mmHg) and duration to cause limited capillary transit and venous filling to assure complete filling of all precapillary vessels and collaterals. After the Microfil had set for 20 min, brains were kept in 4% PFA and collaterals were imaged the next day using a Leica fluorescent stereomicroscope. All collaterals between the anterior cerebral artery (ACA) and middle cerebral artery (MCA) trees of both hemispheres were counted and divided by 2 to give the average number per hemisphere. Images were subsequently analyzed (ImageJ, NIH, Bethesday, MD, USA): Collateral lumen diameter was determined at midpoint at 50X for all ACA-MCA collaterals and averaged for each mouse. 

### 4.2. Permanent Middle Cerebral Artery Occlusion (pMCAO)

As previously described [20], mice were anesthetized with ketamine and xylazine (100 and 10 mg/kg, ip, respectively) and rectal temperature was maintained at 37 ± 0.5 °C. The temporalis muscle between the eye and ear on one side was retracted after a 4 mm incision. After a 2 mm craniotomy (18000-17 drill, FST, Foster City, CA, USA), the dura was incised with a 27 gauge needle tip and reflected to reveal the main trunk of the MCA which was cauterized (18010-00, FST, modified) distal to the lenticulostriate branches. The incision was closed with suture and Vetbond (3M, Minneapolis, MN, USA), intramuscular cefazolin 50 mg/kg and buprenorphine were administered, and the animal was monitored in a warmed cage during recovery from anesthesia to maintain the above rectal temperature. Mice were euthanized 4 days after pMCAO. Pial collateral morphometry was then performed as described above.

### 4.3. Laser Doppler Perfusion Imaging and Hindlimb Ischemia Model

Femoral artery ligation (FAL) was performed and hindlimb perfusion was measured using a laser Doppler perfusion imager (Model LDI2-IR, Moor Instruments, Wilmington, DE, USA, ~2 mm penetration and high resolution) as described previously [71]. Briefly, the anterior thigh and adductor regions were depilated followed by 24 h to recover from any erythema. Mice were then anesthetized with 1.25% isoflurane/O_2_, rectal temperature was maintained at 37 ± 0.5 °C. A 3 mm incision was made overlying the femoral vessels 5 mm proximal to the knee. The femoral artery was gently isolated and ligated twice with 7-0 suture immediately distal to the lateral caudal femoral artery and 1 mm further distally, followed by transection between the ligatures. The superficial epigastric artery was also ligated. The skin was closed using 5-0 silk. Mice were placed in a heated chamber to block ambient light, and rectal temperature was maintained at 37 ± 0.5 °C. Less than 5 min after ligation, the plantar and adductor regions of both legs were laser-scanned. Average perfusion within the region of interest, drawn to outline the plantar surface of the paws, was calculated using Moor LDI Image Processing V5.0 software and reported as the ratio of ligated to the non-ligated hindlimb, where the region of interest was defined according to anatomic landmarks as described previously [71].

### 4.4. Quantitative NanoString Expression Analysis

As described previously [46], mice placed under deep anesthesia (ketamine and xylazine, 100 and 10 mg/kg, ip, respectively) were perfused with 1 ml RNAlater® (Sigma-Aldrich Corp, St. Louis, MO, USA) premixed with 10% Evans blue dye through the thoracic aorta. The brain was then removed and immersed in RNAlater®. Approximately 10 pial collaterals and 10 nearby similarly sized distal-most arterioles per mouse (total 36 mice) were micro-dissected and pooled in RNAlater® as 6 samples. Samples were homogenized (TH, Omni International, Marietta, GA, USA) in Trizol Reagent (Invitrogen, Carlsbad, CA, USA). Total RNA was purified using the RNeasy Micro Kit according to the manufacturer (Qiagen, Valencia, CA, USA). RNA concentration and quality were determined by NanoDrop 1000 (Thermo Scientific, Wilmington, DE, USA) and Bioanalyzer 2100 (Agilent, Foster City, CA, USA), respectively. Measurement of transcript number was conducted for 22 selected genes by the genomics facility at UNC using NanoString custom-synthesized probes (NanoString, Seattle, WA, USA). Transcript number for each gene was normalized to one of the following 6 housekeeping genes selected to be in range of the target gene under analysis: *Gapdh*, *βactin*, *Tubb5*, *Hprt1*, *Ppia* and *Tbp*.

### 4.5. Immunohistochemistry

Mice under deep anesthesia (ketamine and xylazine, 100 and 10 mg/kg, ip, respectively) were perfused with nitroprusside (10^−4^ M) in PBS at ~100 mmHg with a reservoir for 3 minutes for maximal dilation and then with 2% PFA for 15 min for fixation. Brains were blocked with 10% goat serum in 0.3% PBS-triton 1 hr at room temperature. Then the cortex area were incubated with 1:50 anti-eNOS rabbit IgG (sc-654, Santa Cruz Biotechnology, Santa Cruz, CA, USA); 1:100 anti-phospho eNOS rabbit IgG (ab75639, Abcam, Cambridge, MA, USA), 1:100 anti-acetylated-tubulin (T7451, Sigma), 1:100 anti-ZO1 (ab96587, Abcam), 1;400 anti-aSMA (Abcam, 32575, clone E184), overnight, 4 °C in a shaker, followed by incubation of 1:200 goat anti rabbit-fluorescent conjugated with Alexa fluor® 568 (A10042, ThermoFisher Scientific, Grand Island, NY, USA) or Alexa fluor® 488 (A21208). Images of eNOS and phorpho-eNOS for pial collaterals and DMAs were taken using a Leica fluorescent stereomicroscope (Richmond, IL, USA), and the signal strength was measured using ImageJ (Bethesda, MD, USA). Endothelial cells were probed with Isolectin-GS-IB4-Alexa568 (I121415, ThermoFisher Scientific). Primary cilia (probed with anti-acetylated-tubulin antibody) in the lumen of pial collaterals were observed with a Zeiss 710 confocal microscope (Thornwood, NT, USA), with z stack scanning from the top of cortex down to a depth of ~30 um. Endothelial junctions (probed with ZO-1) were visualized with a Zeiss 710 confocal microscope.

### 4.6. Collateral Primary Cilia and Endothelial Orientation Assessed by Scanning Electron Microscopy

Mice were perfused with maximal vessel dilation as described above. Brain arterial vasculature was then casted using a Batson’s No 17 Plastic Replica and Corrosion Kit (Polysciences, Inc, Warrington, PA, USA). Briefly, 1 ml of Batson’s 17 was infused through the thoracic aorta. After fully curing, the brain tissue was removed using maceration solution, and the cerebral vasculature including the pial collateral regions were carefully persevered for emission scanning electron microscopy. The vasculature was observed under a Zeiss Supra 25 Field emission scanning electron microscope. Images of pial collateral and DMAs were saved for analysis of primary cilia and endothelial cell morphology. To measure the orientation of collateral endothelial cells, we use Photoshop to draw a line coordinate with the collateral axis and a second line coordinate with the endothelial cell axis (see Figure 2, panel C), and the angle formed was measured.

### 4.7. Statistics

Experiments were performed in accordance with the University of North Carolina’s Institutional Animal Care and Use Committee, the NIH Guide for the Care and Use of Laboratory Animals, the ARRIVE guidelines, and the following suggested STAIR criteria [72]: investigators were blinded during data analysis where possible; no data points were identified as outliers by statistical test and none were excluded; all results were fully disclosed including negative results; the review, discussion and citation of the literature was unbiased; pMCAO was used to permanently recruit blood flow across pial collaterals. Values are mean ± SEM. Statistical analysis (*p* < 0.05 = significant) is described in the figure legends.

## 5. Conclusions

In conclusion, collaterals are unique among blood vessel types with regard to their formation, structure, function, prevailing shear stress, and susceptibility to variation in their extent caused primarily by differences in genetic background but also by environmental factors such as aging and risk factors. The present study provides our first look into how differences in collateral endothelial and smooth muscle cells may accommodate and contribute to these unique features. Moreover, the model/hypothesis shown in Figure 9, if correct, may begin to provide answers to two perplexing questions: Why do collaterals undergo rarefaction with aging and other vascular risk factors? They are chronically exposed to adverse hemodynamic conditions. How do they resist more extensive pruning away? Their mural cells have specializations or adaptations in structure and expression of factors that mitigate the effects of these conditions.

## Figures and Tables

**Figure 1 ijms-20-03608-f001:**
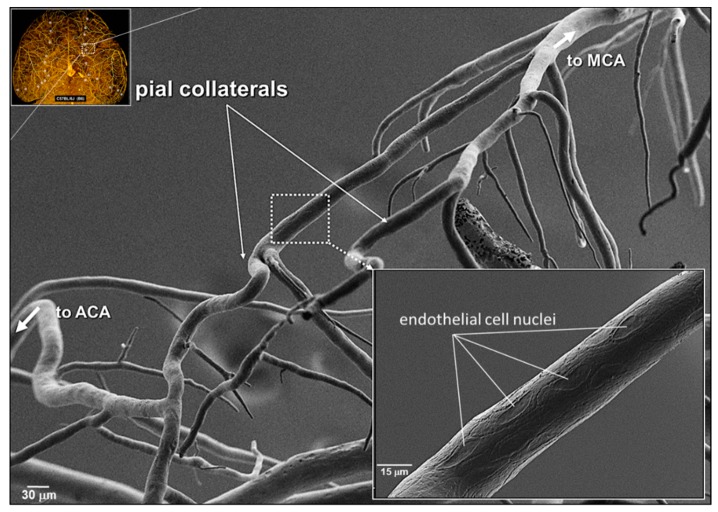
Pial collateral endothelial cells are aligned with the vessel axis. Scanning electron micrograph (SEM) of a corrosion cast of Batson’s #17-filled cerebral pial arterial vessels and 2 collaterals, fixed after maximal dilation, which overlie the watershed zone between the anterior (ACA) and middle (MCA) cerebral artery trees. Upper inset, Microfil^®^ cast of arterial vessels and collaterals (stars) in optically cleared brain. SEMs were obtained from six mice (see also Appendix A). Penetrating arterioles are evident branching from collaterals and pial arterioles.

**Figure 2 ijms-20-03608-f002:**
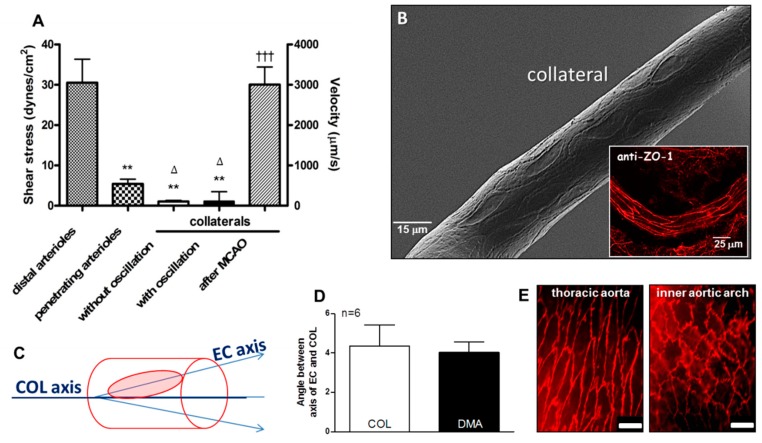
Collateral endothelial cells are aligned with the vessel axis despite having low and oscillatory flow/shear stress in the absence of arterial obstruction. (**A**) Data were obtained in anesthetized mice via cranial window and previously published in reference [6]; unlike distal-most arterioles (DMAs) with diameters comparable to collaterals (COLs) and penetrating arterioles, COLs examined over 30 s intervals have either no flow or slowly oscillating, low velocity, to-and-fro flow with ~zero net-direction. After ligation of the MCA trunk (MCAO), flow to its territory reaches that evident in DMAs within 10–30 s. (**B**–**E**) Collateral endothelial cells (ECs) have the same “anti-inflammatory” alignment (~4 degrees from horizontal) as ECs of DMAs and the descending thoracic aorta, despite having low/disturbed shear stress at baseline. This is in contrast to the “pro-inflammatory” non-alignment present in the inner curvature of the aortic arch. In this and subsequent figures, data are means ± SE for “*n*” number of mice. Data in D determined from SEM images, *n* = 6 mice. Panel E magnification bar is 25 µm. Panel A 2-sided t-tests for shear stress followed by Bonferroni correction for ** *p* < 0.01 vs distal arterioles; ∆ *p* < 0.05 vs penetrating arterioles; 2-sided t-test for ttt *p* < 0.001 vs before MCAO. ZO-1, zona occludens-1 immunohistochemistry.

**Figure 3 ijms-20-03608-f003:**
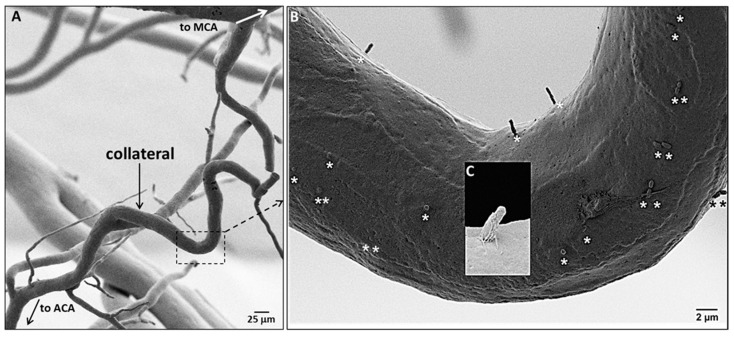
Collateral endothelial cells have primary cilia. (**A**) SEM of a corrosion cast of pial arterial vessels and a collateral, fixed at maximal dilation. (**B**) Stars identify casts of plasmalemmmal invaginations that contain the proximal end of the primary cilia (PrC) filled with Batson’s #17 after removal of the PrC by shearing during infusion of the casting agent; each EC has 0–3 PrC. (**C**) Higher magnification SEM of a ~2 µm long PrC invagination.

**Figure 4 ijms-20-03608-f004:**
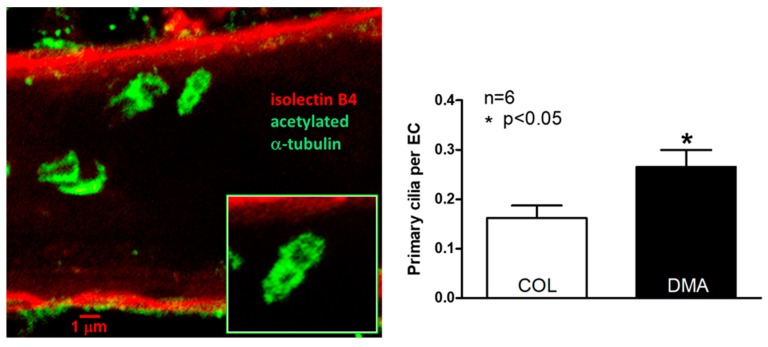
Collateral endothelial cells have fewer primary cilia than distal-most arterioles (DMA). Immunofluorescent-stained collaterals (COL) with focal plane set within the lumen above the far-wall, showing primary cilia. Appendix A shows cilia on DMAs. Inset, higher magnification. Right panel, *n* = 6 mice, 2-sided t-test.

**Figure 5 ijms-20-03608-f005:**
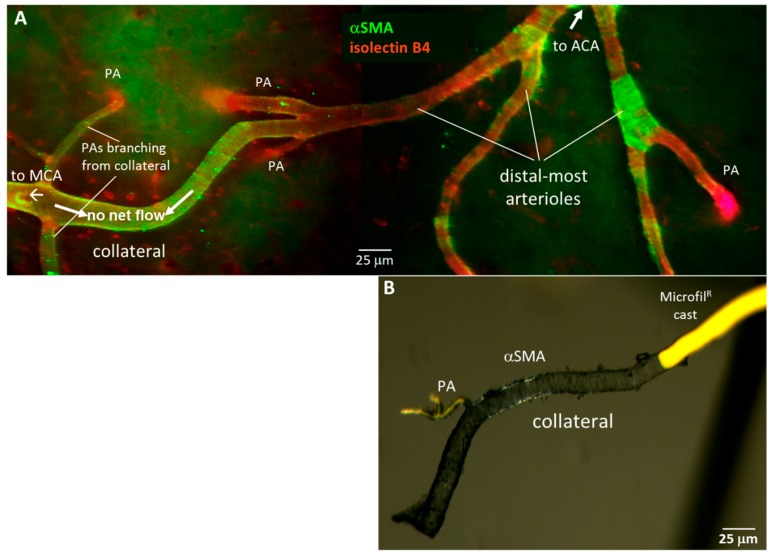
Collaterals are invested with a continuous layer of smooth muscle cells (SMCs), unlike distal-most arterioles (DMAs) that lack or have discontinuous SMCs. (**A**), Immuno-fluorescent staining of SMCs (αSM-actin) and ECs (IB4-lectin). Representative image of pial collaterals, DMAs and penetrating arterioles (PA). (**B**), Brightfield image of αSMA-stained collateral and PA filled with yellow Microfil^®^, then freed from surrounding pial membrane.

**Figure 6 ijms-20-03608-f006:**
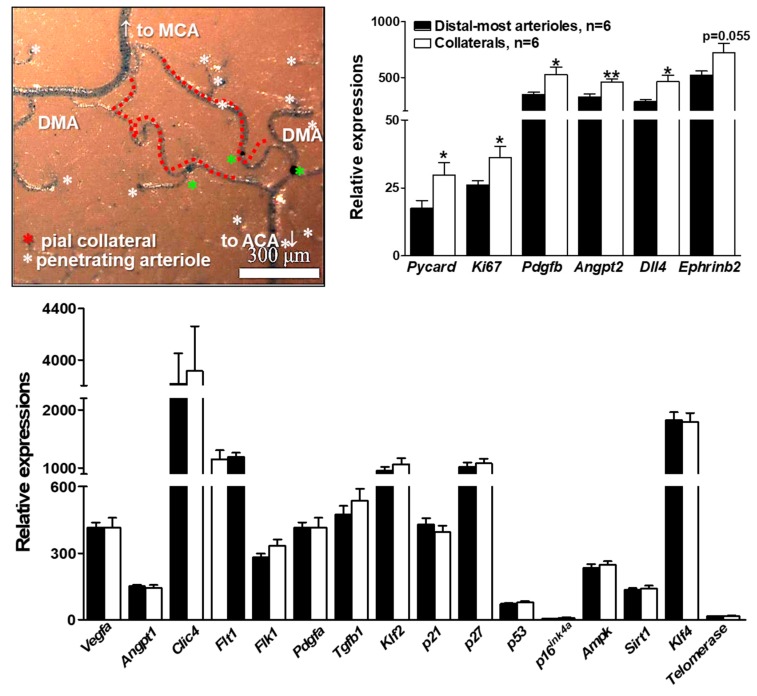
Gene expression differs for collaterals versus distal-most arterioles. Upper left panel, pial arterial vasculature perfusion-fixed at maximal dilation then filled with PU4ii polyurethane. Stars identify penetrating arterioles, including three that bifurcate and descend into the cortex immediately below the arteriole or collateral (green stars). Ten collaterals and 10 nearby similarly-sized distal-most arterioles DMAs were dissected from each of 36 mice and pooled into six samples for extraction of RNA. Transcript abundance was determined by Nanostring n-Counter^®^ for 22 genes, each normalized to one of six housekeeping genes (*Gapdh, βactin, Tubb5, Hprt1, Ppia, Tbp*) selected for comparable level of expression [46]. * *p* < 0.05, ** *p* < 0.01 by 2-sided t-test for collaterals versus DMAs.

**Figure 7 ijms-20-03608-f007:**
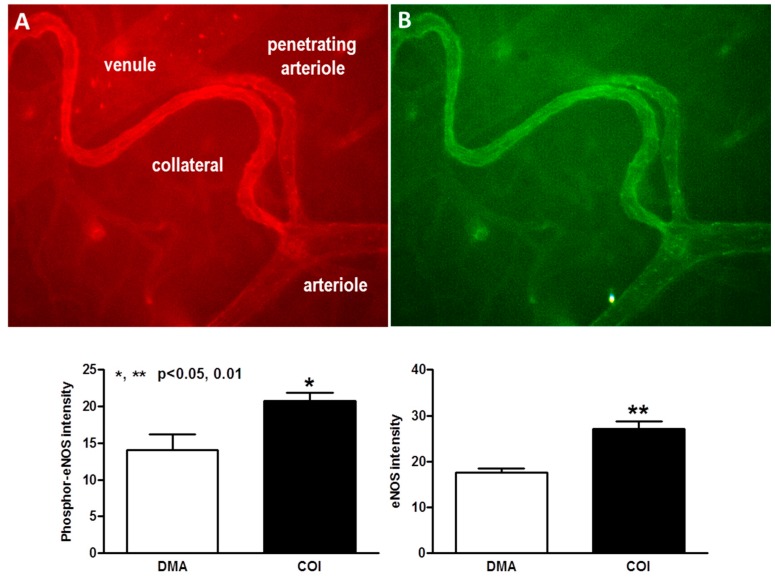
Collaterals express increased levels of phospho- and total eNOS compared to arterioles. Top panels, immunohistochemistry for phospho- (**A**, red) and total (**B**, green) eNOS, *n* = 4 mice, 1-sided t-tests, 163x magnification. * *p* < 0.05, ** *p* < 0.01.

**Figure 8 ijms-20-03608-f008:**
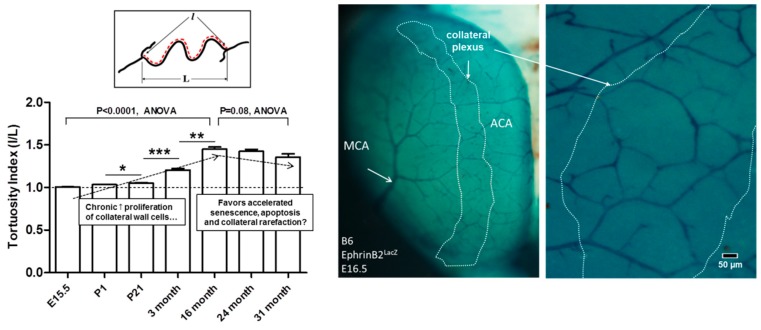
Tortuosity increases progressively with time after formation of collaterals. Images at right, ephrin-B2^LacZ^ reporter mouse (construction of mutant described in [6]) showing embryonic collaterals. Bar graph, tortuosity index = l/L (axial length of collateral ÷ scalar length connecting collateral endpoints). E, embryonic day, P, post-natal day. Data for last four bars from Faber et al. [17] who also showed that collateral diameter and number begin to decline at or after 16 months of age. Number of mice (C57BL/6, B6) for bars 1–7: 8,9,8,9,10,7,8. For each mouse tortuosity, was determined for all MCA–ACA collaterals and averaged. ANOVA followed by 1-sided Bonferroni t-tests, *, **, ***, *p* < 0.05, 0.01, 0.001, respectively.

**Figure 9 ijms-20-03608-f009:**
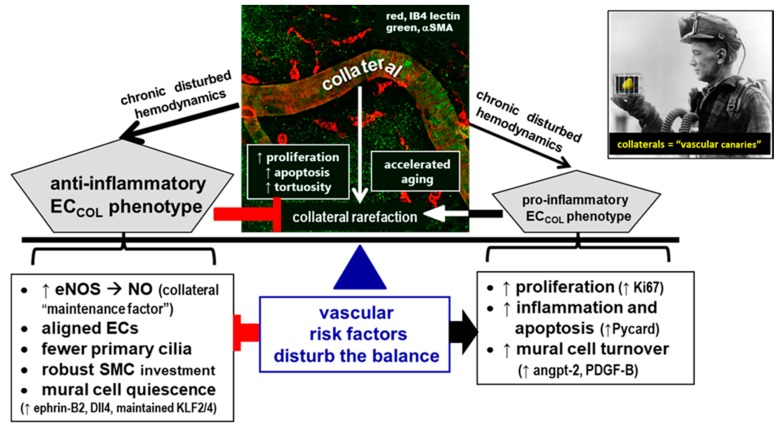
Persistence/maintenance versus rarefaction/pruning of collaterals “hangs in the balance”. Proposed model whereby collateral (COL) mural cells reside in an environment of low/disturbed shear stress, high circumferential wall stress, and low blood oxygen content. This favors a pro-inflammatory, pro-proliferative, pro-apoptotic, and accelerated aging EC phenotype, leading to loss of collateral number and diameter (rarefaction). Compared to distal arterioles, collaterals have specializations and differential gene expression (left box) that provide adaptations that mitigate against factors that promote collateral rarefaction (right box). Vascular risk factors, e.g., aging, hypertension, EC dysfunction, and oxidative stress, disturb the balance. Collaterals are more sensitive than other vessels to these environmental risk factors, like “canaries in a mine-shaft”.

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
