# Peer review of "Collateral Vessels Have Unique Endothelial and Smooth Muscle Cell Phenotypes"

_ijms, 2019, doi:10.3390/ijms20153608_

Reviewer 1 Report

In this study, Zhang et al. compared various phenotypical characteristics of endothelial cells and smooth muscle cells from collaterals to cells from similarly-sized nearby distal-most arterioles. The authors have shown that endothelial cells from collaterals have a different alignment within the vessels and present primary cilia, which are less abundant. Collaterogenesis was not affected by knockdown of ciliary polycystin-1 in endothelial cells. In addition, the authors could show that smooth muscle cells form a continuous layer in collaterals. Also, collaterals had different phospho and total eNOS protein levels as well as transcript levels of few genes involved in proliferation, inflammation and apoptosis. This is a potentially very interesting and exciting finding. However, there are several issues that should be addressed by the authors:

1. The introduction is continued and partly repeated in the results section. The results section should be rewritten to present only the authors own work, while the hypothesis and relevant literature moved to introduction.

2. Along these lines, the interpretation of their observations compared to the knowledge so far should be moved from “results” and integrated in the discussion section.

3. Also, the final conclusion in the abstract is not supported by data: “endothelial and SMCs of collaterals have morphologic and functional specializations that counterbalance the disturbed hemodynamic, pro-inflammatory, pro- proliferative environment in which collaterals reside and thus mitigate their risk factor-induced rarefaction”. That the differences described by the authors “mitigate” any changes should be proven by additional experiments. Otherwise, the conclusions should be rewritten to reflect the findings of the present paper, and not the hypothesis.

4.  Was part of the data presented in Fig. 5 and Fig. 9 already published in other papers, according to the indication in figure legends? It is not clear whether the experiments and the analysis of all the groups were performed simultaneously and the proper controls were used. Please clarify.

5.  Fig. 2A - the use of t-test for statistical analysis of multiple groups is not appropriate. The authors should use ANOVA with posthoc for this type of data. The recalculation of the statistics is required.

6.  The n numbers should appear in the figure legends, not the figure itself.

7.  Was the knockdown efficiency of polycystin-1 verified in these mice? Is polycystin-2 upregulated as an adaptive mechanism in these mice? Could, in this case, polycystin-2 take over the function to compensate the loss of polycystin-1?

8.  The results for gene expression should be shown as graphs for each gene of interest, and not as a table.

9.  The analysis of genes expression in collaterals and DMAs may not be a suitable method to analyse differences between various processes in these two types of vessels. Gene expression does not reflect always the changes at the protein level or function. The investigation of oxidative stress, inflammatory, proliferation or senescence markers at the protein level may reflect better the differences in cellular characteristics. The authors should provide more data on the cellular phenotype to support the hypothesis and the model proposed in Fig. 10. These would strengthen the paper.

Author Response

Thank you for your review and your comment “This is a potentially very interesting and exciting finding.”

1. The Results section has been revised accordingly.  However, there are several places where it was necessary to reference the work of others: 1) to frame, for the Reader, the rationale for the question/experiment at hand and thus aid comprehension of the findings, or 2) to address a minor finding/issue that, if left to be dealt with in Discussion, would require that the rationale for the experiment and findings be restated, thus detracting from concise composition.  These usages in Results are routine in many journals including those with high impact. 

2. We have deleted the last sentence of the second paragraph of Results (“Also, they are much more abundant than reported previously for conduit vessels, wherein cilia are either absent or present on less than one percent of ECs [32,34].”), though we think it aided Reader’s comprehension.  At the end of paragraph 3, we would like to retain the brief interpretation (“The above results suggest that normal/baseline levels of polycystin-1 are not required for collaterogenesis or collateral remodeling.”) for the reasons stated above and in our response to question 1.  For similar reasons, we would also ask that we be permitted to retain the sentence at the end of the last paragraph of Results.  The above are the three instances of what is referenced in question 2, if we understand the Reviewer’s question correctly.

3. The concluding statements in the abstract have been revised accordingly:

In conclusion, endothelial and SMCs of collaterals have morphologic and functional differences from those of nearby similarly sized arterioles.  Future studies are required to determine if they represent specializations that counterbalance the disturbed hemodynamic, pro-inflammatory, pro-proliferative environment in which collaterals reside and thus mitigate their risk factor-induced rarefaction.

4.  Yes.  This is now clarified further in both legends.

5.  As the legend states, Figure 2A, including its indicated statistical tests, is taken from our previous paper whose reference is given therein [6].  The statistical tests were approved by the reviewers of that paper because they were a-priori hypothesized comparisons of only the indicated two groups that were tested, and because multiple group comparisons either had no scientific meaning or were not pre-specified.  The comparisons were also set at 2-sided (as stated in the legend) to be conservative, even though they were pre-specified as one-sided from a known/scientific standpoint (eg, the pre-specified comparison of interest was the effect of MCAO on collateral flow and shear stress, and it was expected to increase after acute MCAO based on hemodynamic principles and many previous publications).

6.  Two of the 10 figures in the paper have n-sizes indicated in the figure’s (or figure panel’s) key, and Figure 9 gives them in the base of the 7 bars that are shown.  We believe in these instances that giving them in this manner, along with the error bars, aids the Reader’s quick assessment of the robustness of the data, rather than having to read through the legend or return to the Methods section.  We request that we be permitted to retain them.

7.  Knockdown efficiency was not tested.  We forgot to state this.  Thank you for the query.  The following is now stated at the end of the paragraph in Discussion that addresses the polycystin-1 results (the new text that has been added is underlined below):

These findings suggest that normal levels of polycystin-1 are not required for collateral formation during development or collateral remodeling after occlusion.  However, potential compensation by other ciliary proteins or possible insufficient knockdown of polycystin-1 weaken this conclusion, as does the small n-size in Figure 6.  However, the fact that the same outcome was obtained in both brain and hindlimb and that collateral diameter was smaller in the knockdown mice—indicating knockdown was achieved (note, we did not measure knockdown of polycystin-1 per se)—mitigate these weaknesses.  Studies using conditional EC-specific knockdown during collaterogenesis and prior to artery ligation in adults are required to confirm these conclusions.  Other approaches to interfere with cilia presence and function, eg, with knockdown of other ciliary proteins such as Pkd2 and Ift88, also need to be examined. 

8.   Changed as requested.

9.   The following has been added as the last sentence of the second to last paragraph in Discussion:

However, analysis of gene transcription does not always reflect changes in protein level or function, thus future work investigating oxidative stress, inflammatory, proliferation and senescence markers at the protein level may better reflect differences in cellular characteristics.

Reviewer 2 Report

p.p1 {margin: 0.0px 0.0px 0.0px 0.0px; font: 12.0px Helvetica} p.p2 {margin: 0.0px 0.0px 0.0px 0.0px; font: 12.0px Helvetica; min-height: 14.0px}

Zhang et. al. presents work on collateral vessels that seeks to identify structural characteristics not found in other vessels. This reviewer found this work to be overall very novel ad should be of interest to the field. Minor comments are noted below:

1.  fig 1 scale bar - cut off

2.  Figure 2A - How was this plot generated and where does it come from? It looks as if it was cut and paste from somewhere else. I would suggest improving image quality of this figure.

3.  Figure 2E - This figure looks blurry and its very difficult to see scale bar. Please use better quality images.

4.  Figure 4 - do the authors have a complimentary immunofluorescence picture of DMA stained for cilia? It would be helpful for the reader toe be able to do a visual comparison as well.

- Also, the plot in this figure is confusing. What does n = 6 (subscript 184) mean?

5.  Figure 8 - it would be helpful if each sub figure in this figure were labeled with letters, 8A, 8B, etc.

- Phospho-eNosintensity figure seems to have some white letter overlayed on the y-axis.

Author Response

Thank you for your review and your comment “This reviewer found this work to be overall very novel and should be of interest to the field.”

1. Thank you - we had not seen the Journal’s reformatted version, which caused this.  It has been corrected.

2. The legend, which indicated that it was previously published, has been revised (the following underlined words were added) “Data were obtained in anesthetized mice via cranial window and previously published in reference [6].  We included this panel to help the Reader understand the context and interpretation of the new data presented in the rest of the figure without Reader having to consult reference 6.  The quality of panel A has been corrected.

3. The scale bar was blurry because it was too small.  It has been corrected.

4. We did not add an immunoblot showing cilia on DMAs to Figure 4 for two reasons: 1) Because it would not be possible to depict with “representative” immunoblots what the bar graph shows, ie, that there are 40% fewer on collaterals, without showing representative immunoblots at very low power; we thought that Readers would prefer a high power image to judge the quality of the staining vis-a-vis primary cilia as shown by others.  2) Because, as expected, the immunostained cilia on DMAs look the same as those on collaterals.  Given these reasons, we thought it more informative to instead provided Supplemental figure S2 which shows, using SEM in large field format, that that cilia with comparable morphology are also on DMAs.  We have added this referenced to Supplemental figure S2 to Figure 4’s legend.  Regarding “n=6”, Figure 4’s legend had stated the following “Right panel, n = 6 mice, 2-sided t-test.”

5.  We had defined the colors in the legend.  We have now also added the letters A and B to the upper two panels and in the legend.  The errant overlay has been removed.

Reviewer 3 Report

The authors assessed the different phenotypes of endothelial and smooth muscle cells in collateral blood vessels and their contribution in remodelling in an animal model.

1. The study is well desinged and the data sufficiently presented.

2. The discussion is lengthy na dmay be shortened.

Author Response

Dear Reviewer,

Thank you for your reviews.  Please see our comments in red font regarding your questions/suggestions/comments.

JEF

1.  Thank you.

2.  Per Reviewer 1’s comment, we have deleted the Pkd1 knockdown results (old Figure 5), resulting in the deletion of text from Discussion thus shortening its length.

Reviewer 4 Report

1.  An excellent and novel paper, opening the way for many future studies. This study raises many questions regarding the development and formation of collaterals. Are they derived from the same stem cell niche and the macro/micro vasculature? What are the exposome cues that epigenetically program theses cells? It would be interesting to see a future integrated 'omics' study to comprehensively define both Col ECs and SMCs. Is impossible to recapitulate these cell types in culture for comprehensive in vitro studies?

2.  The authors may have uploaded a version of the paper displaying corrections and edits. Minor editing required for the manuscript, scientifically and experimentally sound, novel findings well presented, interrogated and interpreted.

3.  This paper will add significantly to our current body of knowledge, and will be of interest to a wide readership.

Author Response

Dear Reviewer,

Thank you for your reviews.  Please see our comments in red font regarding your questions/suggestions/comments.

JEF

1.  Thank you.

2.  We have re-checked the manuscript for edits of any typos etc, and will do so again in the galley version.

3.  Thank you.

Round  2

Reviewer 1 Report

In the revised manuscript, the authors improved their study. However, several important issues still remain to be addressed by the authors:

1. “As the legend states, Figure 2A, including its indicated statistical tests, is taken from our previous paper whose reference is given therein [6].  The statistical tests were approved by the reviewers of that paper because they were a-priori hypothesized comparisons of only the indicated two groups that were tested, and because multiple group comparisons either had no scientific meaning or were not pre-specified.  The comparisons were also set at 2-sided (as stated in the legend) to be conservative, even though they were pre-specified as one-sided from a known/scientific standpoint (eg, the pre-specified comparison of interest was the effect of MCAO on collateral flow and shear stress, and it was expected to increase after acute MCAO based on hemodynamic principles and many previous publications).”

The t-test can be used for multiple comparisons, but in this case the alpha adjustment should be performed. Was it performed in this case? If it is not wished by the authors, one could just remove the graph and simply cite the findings from the previous publication.

2. “Two of the 10 figures in the paper have n-sizes indicated in the figure’s (or figure panel’s) key, and Figure 9 gives them in the base of the 7 bars that are shown.  We believe in these instances that giving them in this manner, along with the error bars, aids the Reader’s quick assessment of the robustness of the data, rather than having to read through the legend or return to the Methods section.  We request that we be permitted to retain them.”

If the author’s wish is to aid the readers in quickly visualizing the robustness of the data, the authors should present the data as scatter dot blots and not bar graphs, a requirement for presenting data in many journals, especially in the high impact journals.

3. “Knockdown efficiency was not tested.  We forgot to state this.  Thank you for the query.  The following is now stated at the end of the paragraph in Discussion that addresses the polycystin-1 results (the new text that has been added is underlined below): These findings suggest that normal levels of polycystin-1 are not required for collateral formation during development or collateral remodeling after occlusion.  However, potential compensation by other ciliary proteins or possible insufficient knockdown of polycystin-1 weaken this conclusion, as does the small n-size in Figure 6.  However, the fact that the same outcome was obtained in both brain and hindlimb and that collateral diameter was smaller in the knockdown mice—indicating knockdown was achieved (note, we did not measure knockdown of polycystin-1 per se)—mitigate these weaknesses.  Studies using conditional EC-specific knockdown during collaterogenesis and prior to artery ligation in adults are required to confirm these conclusions.  Other approaches to interfere with cilia presence and function, eg, with knockdown of other ciliary proteins such as Pkd2 and Ift88, also need to be examined.”

No proof for knockdown efficiency of polycystin-1 raises major concerns regarding the data. The claim that polycystin-1 knockdown has no effects on collateral formation without exploring whether the knockdown was actually successful is not acceptable. The authors should remove all data that is not generated from confirmed experimental models. Presenting the data as a clear finding, and just suggesting in the discussion that the knockdown could be insufficient, without any proof that it worked at all, is misleading for the readers. The experimental design has a major flaw.

4. Additional experiments (as suggested for the first version of the manuscript) would strengthen the paper.

Author Response

Dear Reviewer,

Thank you for your reviews.  Please see our comments in red font regarding your questions/suggestions/comments.

JEF

1. The data have been re-tested with Bonferroni-corrected alpha values, and the symbols on the figure and figure legend corrected.

2. As originally requested by the reviewer, all n-values have been removed from the figures and placed in the legends.

3. We no longer have the mouse lines needed to examine KD efficiency, thus the figure has been removed and the text of the manuscript revised accordingly.

4. We agree.  We have pointed out limitations of our study in the Discussion and in the second to last paragraph have identified several areas of future investigation.  We have also modified the polycystin-1 paragraph in Discussion to point out future studies that are needed to determine if collateral formation and remodeling require functions provided by primary cilia on the collateral ECs.